# The Aftermath of COVID-19: Exploring the Long-Term Effects on Organ Systems

**DOI:** 10.3390/biomedicines12040913

**Published:** 2024-04-20

**Authors:** Maryam Golzardi, Altijana Hromić-Jahjefendić, Jasmin Šutković, Orkun Aydin, Pinar Ünal-Aydın, Tea Bećirević, Elrashdy M. Redwan, Alberto Rubio-Casillas, Vladimir N. Uversky

**Affiliations:** 1Department of Genetics and Bioengineering, Faculty of Engineering and Natural Sciences, International University of Sarajevo, Hrasnicka Cesta 15, 71000 Sarajevo, Bosnia and Herzegovina; maryam.golezardi@gmail.com (M.G.); jsutkovic@ius.edu.ba (J.Š.); 2Department of Psychology, Faculty of Arts and Social Sciences, International University of Sarajevo, Hrasnicka Cesta 15, 71000 Sarajevo, Bosnia and Herzegovina; oaydin@ius.edu.ba (O.A.); paydin@ius.edu.ba (P.Ü.-A.); 3Atrijum Polyclinic, 71000 Sarajevo, Bosnia and Herzegovina; becirevic.tea@gmail.com; 4Department of Biological Science, Faculty of Science, King Abdulaziz University, Jeddah 21589, Saudi Arabia; lradwan@kau.edu.sa; 5Centre of Excellence in Bionanoscience Research, King Abdulaziz University, Jeddah 21589, Saudi Arabia; 6Therapeutic and Protective Proteins Laboratory, Protein Research Department, Genetic Engineering and Biotechnology Research Institute, City of Scientific Research and Technological Applications (SRTA-City), New Borg EL-Arab, Alexandria 21934, Egypt; 7Autlan Regional Hospital, Health Secretariat, Autlan 48900, Jalisco, Mexico; alberto.rubio@sems.udg.mx; 8Biology Laboratory, Autlan Regional Preparatory School, University of Guadalajara, Autlan 48900, Jalisco, Mexico; 9Department of Molecular Medicine and USF Health Byrd Alzheimer’s Research Institute, Morsani College of Medicine, University of South Florida, Tampa, FL 33612, USA

**Keywords:** SARS-CoV-2, long-COVID, organ systems

## Abstract

Background: Post-acute sequelae of SARS-CoV-2 infection (PASC) is a complicated disease that affects millions of people all over the world. Previous studies have shown that PASC impacts 10% of SARS-CoV-2 infected patients of which 50–70% are hospitalised. It has also been shown that 10–12% of those vaccinated against COVID-19 were affected by PASC and its complications. The severity and the later development of PASC symptoms are positively associated with the early intensity of the infection. Results: The generated health complications caused by PASC involve a vast variety of organ systems. Patients affected by PASC have been diagnosed with neuropsychiatric and neurological symptoms. The cardiovascular system also has been involved and several diseases such as myocarditis, pericarditis, and coronary artery diseases were reported. Chronic hematological problems such as thrombotic endothelialitis and hypercoagulability were described as conditions that could increase the risk of clotting disorders and coagulopathy in PASC patients. Chest pain, breathlessness, and cough in PASC patients were associated with the respiratory system in long-COVID causing respiratory distress syndrome. The observed immune complications were notable, involving several diseases. The renal system also was impacted, which resulted in raising the risk of diseases such as thrombotic issues, fibrosis, and sepsis. Endocrine gland malfunction can lead to diabetes, thyroiditis, and male infertility. Symptoms such as diarrhea, nausea, loss of appetite, and taste were also among reported observations due to several gastrointestinal disorders. Skin abnormalities might be an indication of infection and long-term implications such as persistent cutaneous complaints linked to PASC. Conclusions: Long-COVID is a multidimensional syndrome with considerable public health implications, affecting several physiological systems and demanding thorough medical therapy, and more study to address its underlying causes and long-term effects is needed.

## 1. Introduction

SARS-CoV-2 (severe acute respiratory syndrome-coronavirus-2) is a fatal coronavirus that appeared in late 2019. It generated the global pandemic of coronavirus disease 2019 (COVID-19), an acute respiratory and systemic illness that endangers public health globally [1]. Since the outbreak in 2019, the COVID-19 pandemic has been the centrepiece for studies on the signalosome [2], therapeutic treatments [3], and its potential role in other illnesses ([4,5]). Thousands of SARS-CoV-2 patients are suffering from a variety of post-COVID problems, including so-called “long-COVID”.

Long-COVID, also known as post-acute sequelae of SARS-CoV-2 infection (PASC) is a multisystemic sickness with frequent significant signs that arise following an acute respiratory syndrome coronavirus 2 (SARS-CoV-2) [6]. According to a cautiously anticipated prevalence, with over 651 million reported COVID-19 cases worldwide, at least 10% of infected individuals have experienced long-COVID [6]. Studies have reported that 10–30% of non-hospitalised patients and 50–70% of hospitalised patients are impacted by PASC [7]. Interestingly, it has been described that 10–12% of people who have received vaccinations have faced long-COVID [8]. Long-term COVID symptoms include brain fog, tiredness, insomnia, arthralgia, myalgia, pharyngitis, headaches, fever, digestive issues, skin rashes, and emotional symptoms such as sadness and anxiety, which are analogous to those reported during the early phase of infection [9,10,11]. The severity of long-COVID symptoms was shown to vary in different patients and this contrast was observed to be associated with the degree of the early infection [12]. Several pathways have been proposed for protracted COVID-19 pathogenesis, including immunological dysregulation, microbiome disruption, autoimmune, clotting, and endothelial abnormalities, and faulty neurological signaling [13]. The effect of long-COVID exceeds the individual limit and impacts society as significant numbers of patients with long-COVID were reported to be unable to return to work [14], which contributes to labour shortages.

Long-COVID is likely to have numerous causes, some of which may overlap. Several hypotheses have been proposed for its pathophysiology, including persistent reservoirs of SARS-CoV-2 in tissues [15], immune deregulation [16], reactivation of underlying infections, including herpesviruses like Epstein–Barr virus (EBV) and human herpesvirus 6 (HHV-6), among others [17,18], impacts of SARS-CoV-2 on the microbiota (including the virome) [19,20], autoimmunity [21], priming of the immune system from molecular mimicry [20], and microvascular blood clotting with endothelial dysfunction [22]. The summary of organ systems affected by long-COVID is represented in Figure 1.

For a long period of time, researchers tried to observe the symptoms and manifestation of long-COVID on various organ systems. It is known that long-COVID participates in many organ systems including the central nervous system (CNS), gastrointestinal system (GI), respiratory system, renal system, cardiovascular system (CVS), endocrine system, immunological system, and on the skin. This review article seeks to thoroughly explore the emerging evidence and present understanding of the intricate interplay between various organ systems and long-term COVID-19 manifestations, shedding light on the potential implications on different physiological systems causing persistent symptoms.

## 2. Long-COVID and the Nervous System

Long-COVID is linked to a wide range of neuropsychiatric and neurological symptoms that may significantly disrupt daily activities, posing a significant public health issue [23,24,25,26].

The neurological and neuropsychiatric symptoms included in this category are cognitive impairment (brain fog), memory issues, ageusia (loss of taste), anosmia (loss of smell), headaches, sleep disturbances, depression, and anxiety [23,24,26]. Based on the clinical case definition of long-COVID developed by the World Health Organization (WHO) using the Delphi methodology, the prevalence of all symptoms, including neuropsychiatric symptoms, was higher in hospitalised patients compared to non-hospitalised patients (53% vs. 35%). Additionally, the prevalence was higher in women compared to men (49% vs. 37%). The predominant neuropsychiatric symptom reported was brain fog, with a prevalence of 74%. This was followed by memory impairment (65%), sleep difficulties (62%), changed sense of smell (hyposmia) or taste (hypogeusia) (57%), headache (56%), and depression (50%) [27].

COVID-19 infection of the respiratory system leads to inflammation of the neurological system via systemic chemokines and increased microglial activity. The dysregulation of cytokines, chemokines, and reactive microglia in the central nervous system disrupts many kinds of neural cells. It also negatively affects the balance and adaptability of myelin which leads to dysregulation of blood–brain barrier (BBB). In addition, it inhibits neurogenesis in the hippocampus and triggers harmful astrocyte reactivity, which causes the death of oligodendrocyte cells and vulnerable neurons. Hence, COVID-19 infection may have an adverse impact on the functioning of neural circuits and therefore cause cognitive impairment [28].

One of the most common (20–35%) [23] and most distressing neurological effects of long-COVID is a syndrome called “brain fog”, which refers to a chronic cognitive impairment and was even observed in those who had a moderate SARS-CoV-2 infection. This condition is marked by difficulties in memory, concentration, information processing speed, attention, and executive function [7,26,28,29,30,31,32]. A study conducted in Italy during the spring of 2020 examined patients hospitalised with COVID-19 pneumonia. The study found that 78% of these patients experienced cognitive impairment in at least one area three months after their infection. The most common cognitive impairments observed were related to executive function (50%), attention and information processing speed (33%), working memory (24%), and verbal memory (10%) [33].

Another recent neuropsychometric study conducted in a New York City hospital tracked patients with mild, moderate, or severe COVID-19 from spring 2020 to spring 2021. The study revealed that seven months after infection, there were noticeable declines in attention (10%), processing speed (18%), memory encoding (24%), and executive function (16%) among the patients [34]. Longitudinal research performed in Spain aimed to assess neuropsychiatric symptoms in individuals with long-COVID. The study reported that 46.8% of patients had cognitive impairment after one year [35].

Additional potential processes that may contribute to neuropsychiatric symptoms associated with COVID-19 include the following: SARS-CoV-2 seldom has the ability to directly invade the neurological system; it has the potential to trigger an immune response that targets the neurological system, leading to an autoimmune reaction; reactivation of dormant herpesviruses, such as the Epstein–Barr virus, might potentially initiate neuropathological conditions; cerebrovascular and thrombotic illness may impede blood circulation, impair the function of the blood–brain barrier, and exacerbate neuroinflammation and ischaemia in neural cells. The severe form of COVID-19 may lead to pulmonary and multi-organ malfunction resulting in hypoxemia, hypotension, and disruptions in metabolic processes. These conditions can also have detrimental effects on neuronal cells [28].

Moreover, findings from 1-year follow-up research indicate that being female and experiencing anxiety and depression at the 4-month point are strong indicators of developing anxiety and depressive symptoms after 12 months [36].

Further longitudinal research conducted in Spain on assessing the neuropsychiatric symptoms in individuals with long-COVID revealed that 45% of participants had mental morbidity, including anxiety, depression, and post-traumatic stress disorder, after one year [35].

Taquet et al. performed a retrospective cohort research that included 236,379 individuals. The study found that 17.4% of the patients had anxiety disorder and 13.7% had a mood disorder within 6 months of being diagnosed with COVID-19 [37]. Besides the neuropathological pathways, environmental and lifestyle disruptions, together with the global pandemic, are probable factors that have contributed to an overall decline in mental health. These disruptions include isolation, limited healthcare access, and significant and prolonged changes in daily living on a large scale [23].

Additional frequently noticed neurological symptoms associated with long-COVID include sleep disturbances, with a prevalence of 26%, which have a bidirectional association with mental health problems [38,39,40]. The reported sleep disturbances include trouble sleeping, nightmares, and lucid dreaming [41].

Sensory impairments, such as ageusia or dysgeusia and anosmia, have been documented as a symptom linked to long-COVID. Interestingly, the incidence of these impairments did not vary based on the severity of the infection [23,42,43]. A recent meta-analysis study performed by Trott et al. described that around 12.2% of patients have anosmia and 11.7% suffer from ageusia, which lasts for more than 12 weeks after contracting COVID-19 [44]. Another study conducted in Poland involving 2218 patients who had recovered from COVID-19 revealed that 98 patients (4.4%) experienced smell and taste disorders for up to 3 months after their infection [45]. The etiology of these symptoms remains unknown; nevertheless, it is possible that inflammation and infection-induced immune system dysfunction may serve as the underlying mechanism [23]. The summary of neurological complications can be seen in Table 1.

The ongoing neurological symptoms and neuropsychiatric manifestations of COVID-19 pose a significant global health challenge, with long-lasting consequences potentially causing incapacitating effects. These symptoms can hinder resuming a normal lifestyle and potentially strain healthcare systems worldwide. Efficient treatments have been challenging to find in the first years of the epidemic. Identifying long-term neurological symptoms is crucial for perceptive medical practitioners, as it can guide clinical diagnosis and treatment, reducing unnecessary testing. Further research is expected to improve patient outcomes and satisfaction. It is also essential to monitor the cognitive and neuropsychological abilities of those recovering from COVID-19 [23,46].

## 3. Long-COVID and Cardiovascular System

The most prevalent cardiovascular (CV) symptoms reported in PACS are chest discomfort or tightness, palpitations, dizziness, and a rise in resting heart rate [47]. However, the underlying pathophysiological relationship between PACS and the CV system has not been conclusively identified. Several CV disorders may be linked. The summary of cardiovascular issues can be found in Table 1.

### 3.1. Myocarditis

A study of 29 previously hospitalised COVID-19 patients with unexplained increased troponin levels found that 45% had late gadolinium enhancement (LGE), which is suggestive of myocarditis-like patterns on cardiac magnetic resonance imaging (CMR) about 27 days after discharge. Despite this, individuals did not have elevated C-reactive protein (CRP) or troponin levels, and their left ventricular ejection fraction (LVEF) was normal with no wall motion abnormalities. However, no myocardial strain data were supplied [48].

In another research study based on Ohio State University athletes, 15% had CMR data that met the modified Lake Louise criteria for myocarditis diagnosis, which included myocardial edema and non-ischemic myocardial damage markers. This study received media attention, yet it lacked a control group and included patients lacking myocarditis characteristics [49].

Huang et al. [50] evaluated 26 COVID-19-recovered patients with residual cardiovascular symptoms and discovered abnormalities on CMR in 58% without using the new Lake Louise criteria. Notably, the control group’s abnormality % was unknown, and there were no significant variations in troponin, brain natriuretic peptide (BNP), or CRP levels between individuals with normal and abnormal CMRs. Furthermore, several symptomatic individuals had normal CMR values.

Moulson et al. [54] carried out a large-scale study of 19,378 college athletes who recovered from COVID-19. Seventeen percent (17%) tested positive for the virus. The majority had comprehensive cardiac screening, including electrocardiogram, transthoracic echocardiogram, and troponin level, with CMR conducted as needed. Only 3.0% of individuals tested with CMR had pathology related to COVID-19. However, a later analysis found that the existence of cardiovascular symptoms during illness or exercise, or aberrant findings in the first triad testing, significantly enhanced the risk of abnormal CMR outcomes. The authors recommended a progressive screening technique based on symptom intensity and aberrant triad test findings, which they projected would discover 82% of athletes with myocardial or pericardial involvement caused by COVID-19. A prospective study of 149 healthcare professionals in a non-athlete population found no changes in CMR features, troponin levels, or N-terminal pro-BNP at 6 months post-infection compared to age, sex, and ethnicity-matched seronegative controls [51]. The study cohort had limited comorbidities, with only one patient having severe COVID-19.

### 3.2. Pericarditis

The majority of pericarditis cases in the general population are idiopathic, yet it is widely believed that these instances are the result of viral infections [124]. One research study found that 12% of hospitalised COVID-19 patients had diffuse acute ST alterations compatible with pericarditis [52]. Puntmann et al. found that 20% of patients had a pericardial effusion > 1 cm on CMR versus 7% in risk factor-matched controls; however, no patients experienced symptoms [53]. Moulson et al. [54] and Kotecha et al. [125] both observed a 5% incidence of pericardial effusion, which was usually modest in size.

### 3.3. Coronary Artery Disease

Approximately 20–30% of individuals hospitalised with COVID-19 will have elevated troponin levels, most commonly due to type 2 myocardial infarction [55]. Type 2 myocardial infarction should not be neglected once the acute sickness has passed. Troponin levels above the 99th percentile in COVID-19 patients were linked to a 3–6 times higher risk of coronary artery disease (CAD) [56]. Preliminary research by Nai Fovino et al. demonstrated a tendency for greater high-sensitivity troponin peak during COVID-19 admission in patients with a coronary artery calcium score > 400 than those with a score < 400 (1424 versus 419 ng/L, *p* = 0.084) [57]. Interestingly, patients with COVID-19 had a threefold greater chance of a significant adverse cardiac event assessed at a median of 5 months post-discharge compared to age, gender, and risk factor-matched controls; however, these findings deserve more investigation [58].

## 4. Long-COVID and Hematologic Problems

Hematological disorders were observed to occur following the COVID-19 infection [126]. Different alternations in blood composition can be observed in COVID-19 patients, serving as potential indicators for prognosis and treatment. While most individuals infected with COVID-19 experience a return to normal hematological levels shortly after onset, a small percentage may continue to exhibit elevated parameters, which lead to blood disorders, for an extended period of time, spanning months to years [127]. The summary of hematological symptoms can be found in Table 1.

During and after post-COVID-19 infection, various pathophysiological mechanisms can cause long-term hematological conditions, such as persistent thrombotic endothelialitis and systemic hypercoagulability, leading to the formation of fibrinaloid microclots, platelet hyperactivation, and endothelial dysfunction, and ultimately contributing to a range of clotting disorders [59]. One such disorder is COVID-19-induced coagulopathy, which is characterised by an abnormal immune response leading to both blood clotting and bleeding events [60,61]. Elevated levels of D-dimer and fibrinogen, along with mild thrombocytopenia, modest prolongation of prothrombin time, and activated partial thromboplastin time, are important markers of coagulopathy that have been frequently observed in severely ill COVID-19 patients [60]. Late-onset thrombocytopenia is a condition where the immune system becomes deregulated with time [128,129]. Detailed mechanisms through which SARS-CoV-2 infection leads to immune thrombocytopenia are not completely understood and need additional investigation. However, potential mechanisms underlying SARS-CoV-2-mediated immune thrombocytopenia include immune system dysregulation, molecular mimicry, epitope spreading, and susceptibility mutations in the suppressor of the cytokine signaling 1 gene [126]. A systematic analysis revealed that out of 45 individuals with immune thrombocytopenia caused by COVID-19, 20% experienced thrombocytopenia three weeks after the initial COVID-19 symptoms appeared, while 9% encountered a relapse after responding to treatment during the monitoring period [62].

It is already known that during a severe SARS-CoV-2 infection, a cytokine storm occurs, with significantly elevated levels of interleukins (primarily IL-6, IL-2, IL-7, granulocyte colony-stimulating factor, interferon-γ inducible protein 10, MCP-1, and MIP1-a) and tumor necrosis factor (TNF)-α, which can lead to lymphocyte apoptosis [126]. It is noted that persistent lymphocytopenia may still be present even after 5 weeks of the onset of the disease, particularly in patients with severe acute COVID-19. A study conducted in China on 435 hospitalised COVID-19 patients revealed that lymphocytopenia was especially notable in CD3+, CD4+, CD8+, CD19+, and CD16/56+ lymphocyte subsets [63].

A prospective cohort study on coagulation was conducted in 2023, with 102 individuals who had recovered from COVID-19. The results showed that 75% of the patients had a procoagulant state at the 3-month follow-up, while the percentages decreased to 50% at 6 months and 30% at 12–18 months. This association between persistent symptoms and a procoagulant state suggests that ongoing thrombi formation and/or persistent micro thrombosis may be responsible for the main physical symptoms experienced by long-COVID patients [61].

## 5. Long-COVID and Respiratory System

COVID-19 primarily impacts the respiratory system, causing a wide array of clinical and radiological indications. While around 80% of cases involve infection in the upper airways, in 20% of cases, the virus progresses to the alveoli, leading to the development of pulmonary infiltrates [130]. Numerous studies have documented a range of persistent respiratory symptoms in individuals who have recovered from acute COVID-19. The summary of respiratory symptoms can be found in Table 1. PASC is mainly characterised by respiratory symptoms such as chest pain, breathlessness, and cough. Scientific research has validated that even 1 to 3 months post-discharge, patients continued to exhibit radiological abnormalities indicative of pulmonary dysfunction, a reduction in the diffusion capacity for carbon monoxide, and weakened respiratory muscle strength [64,65,66,67]. However, these symptoms tend to decrease over time, with abnormalities in lung function or chest imaging being less common after 12 months compared to 6 months post-discharge [131].

In COVID-19-infected patients, interstitial pneumonia is the primary reason for hospitalisation, with the majority of cases being classified as mild to moderate. Nevertheless, approximately 5–10% of individuals experience a progression to severe respiratory failure and acute respiratory distress syndrome (ARDS) [132,133,134]. Even after recovering from COVID-19, many patients continue to suffer from respiratory symptoms, and multiple studies have indicated abnormalities in pulmonary function tests (PFTs) and chest CT images, even months after their hospital admission. The prevalence of these findings varies, influenced by the approach taken in the research and the duration of follow-up [135,136].

A large meta-analysis study conducted by Long et al. [68], where a total of 4478 COVID-19 patients from 16 cohort studies were included, showed that the prevalence of abnormalities in lung function was approximately 20%. The most common abnormality observed was diffusion impairment, followed by restrictive ventilatory defects [68]. Similar findings have been reported by other studies, such as, for example, the study that evaluated a total of 1200 patients, with 83 suffering from post-COVID-19 pulmonary complications and exhibiting lung function abnormalities, emphasising the potential long-term consequences on respiratory function [137].

Further, patients who were not hospitalised during the acute phase of the COVID-19 infection showed abnormal breathing patterns and respiratory movements, chest pain, reduced lung volumes, air flow, reduced respiratory muscle strength, physical capacity, and thoracic expansion may be involved in long-COVID [138]. In 2023, a cohort study measured the diffusion capacity (DLCO) of non-hospitalised and hospitalised patients at the University Hospitals of Umeå and Örebro, and Karlstad Central Hospital, Sweden. The measured DLCO among hospitalised patients was reduced by 20%, compared to non-hospitalised patients, which was reduced by 4%, indicating a significant difference among the analysed patients [69].

## 6. Long-COVID and Renal System

COVID-19 may influence the kidney in a variety of ways, with the contribution of these variables changing over time. Many of the SARS-CoV-2 virus’s direct and indirect effects may linger beyond hospital release, increasing the risk of sepsis, recurrent acute kidney injury (AKI), and chronic kidney disease (CKD). The summary of renal complications can be found in Table 1. Furthermore, a new diagnosis of diabetes mellitus and deteriorating cardiovascular disease owing to COVID-19 may prolong AKI recovery. Second, the association between COVID-19 and CKD is likely to be bidirectional. Mild CKD may increase the risk of COVID-19 and related AKI, whereas severe AKI may be linked with chronic renal dysfunction, delayed recovery, and/or the requirement for long-term dialysis [70].

US research that used electronic health data from the Veterans Health Administration to perform a complete assessment of long-COVID found that COVID-19 increased the risk of CKD, with the largest risk among individuals with severe disease [71]. Patients hospitalised with COVID-19 had a comparable risk of CKD to those hospitalised with influenza.

The findings of Veterans Health Administration research were comparable to the results of a study based in China on COVID-19 patients, which indicated that 35% of patients had impaired kidney function (6 months after hospitalisation) (estimated glomerular filtration rate (eGFR)  < 90 mL/min/1.73 m^2^). Additionally, 13% of patients who did not have AKI during hospitalisation had a decrease in eGFR at follow-up [38].

COVID-19 pathophysiological pathways may be accentuated in conjunction with deteriorating diabetes or hypertension control. Kidney disease development in COVID-19 is most likely complex and may be triggered by persistent inflammation, intrinsic tubular damage, or maladaptive repair among other mechanisms. Novel kidney-specific plasma and urine biomarkers may aid in determining the primary underlying causes and predicting which individuals are most likely to develop CKD during COVID-19 hospitalisation [139].

A specific interaction between a receptor called angiotensin-converting enzyme 2 (ACE2) and the spike protein of SARS-CoV-2 in different tissues has been identified [140]. It is well known that SARS-CoV-2 gains cell entry through ACE2. When it comes to the assessment of the mechanism, ACE2 is essential for regulating the cardiovascular and renal systems via the Renin–Angiotensin–Aldosterone System (RAAS). Renal juxtaglomerular cells start releasing renin in response to certain situations, such as decreased renal perfusion, hypotension, ischaemia, salt diuresis, and sympathetic stimulation, which changes liver-derived angiotensinogen to angiotensin I (ANGI) [72]. ACE2 then converts ANGI to ANGII, which is further metabolised by ACE2 into ANG I-7. In cardiorenal illness patients, ANGII can modulate fibrosis, hypertrophy, and pro-oxidative hormone, resulting in increased salt and water retention [73]. SARS-CoV-2 reaches the kidneys via the glomerular capillaries and arteries, where it infects the podocytes. SARS-CoV-2 penetrates tubular fluid and eventually reaches proximal tubule epithelial cells [72]. SARS-CoV-2-induced ACE2 receptor internalisation causes ACE2 insufficiency in cells, shifting the ANG I-7-Mas pathways to angiotensin II–angiotensin II type 1 receptor (ANGII-AT1R) and inducing a proinflammatory state [141].

Furthermore, ACE2 controls the Kallikrein Kinin System. In this context, after generating kinin from bradykininogens, this metabolite works on both B1 and B2 receptors to cause inflammation and enhance vascular permeability [74]. Furthermore, Kallikrein can cause a coagulation imbalance by activating factor 12 and plasmin. The renal alterations may increase systemic inflammation, thrombotic issues, fibrosis, and necrosis caused by COVID-19 in the kidneys, culminating in renal impairment [74].

Ramamoorthy et al. [75] reported for the first time that renal fibrosis might occur in the long term after infection. The observation of this study shows elevated mRNA levels of transforming growth factor-beta 1 (TGF-β1), fibroblast growth factor 23 (FGF23), neutrophil gelatinase-associated lipocalin (NGAL), interleukin 18 (IL-18), hypoxia-inducible factor 1-alpha (HIF1-α), Toll-like receptor 2 (TLR2), chitinase-3-like protein 1 (YKL-40), and β2 microglobulin (B2M) long after MHV-1 infection. Long-term post-MHV-1 infection was associated with increased protein levels of HIF1-α, TLR-2, and EGFR, as revealed by immunoblot investigations. Only keyboard input monitor 1 (KIM-1) and matrix metalloproteinase 7 (MMP-7) protein levels, as well as NGAL mRNA, rise during acute infection (7 days). Treatment of MHV-1-infected mice with a synthetic peptide, SPIKENET (SPK), which inhibits spike protein binding, lowered NGAL mRNA in acute infection, and decreased TGF-β1, B-cell CLL/lymphoma 3 (BCL3) mRNA, as well as epidermal growth factor receptor (EGFR), HIF1-α, and TLR-2 protein levels long-term post-infection. These data clearly show that renal fibrosis may occur in the long-term following infection. As a result, tackling these parameters may lead the way to avoiding long-term COVID-19 problems [75].

## 7. Long-COVID and Immune Complications

Oxidative stress and hyper-inflammation emerge as crucial elements in the pathogenesis of COVID-19 and long-COVID syndrome [142]. The impact of free radicals is associated with the progression of cardiovascular, neurodegenerative, and endocrine diseases, inflammatory conditions, and infections, including COVID-19. Oxidative stress plays a crucial role in COVID-19, acting as a significant cause of harm through reactive oxygen species (ROS) and serving as a vital indicator of disease severity. In the acute phase of COVID-19, the observed variations in ROS and oxidative stress levels suggest significant structural alterations in macromolecules, the formation of non-functional derivatives, and substantial harm to cellular components in moderate and critical cases of COVID-19 [143]. Other studies have shown that the immune response against COVID-19 infection is connected to the adaptive immunity [76,144]. Today, long-COVID-related individuals exhibit systemic inflammation and immune dysregulation. As evidence, the global variations in the distribution of CD4+ T and CD8+ T in human cells indicate ongoing immune responses [76]. Patients also display higher levels of SARS-CoV-2 antibodies and a lack of coordination between their SARS-CoV-2-specific T and B cell responses. These findings suggest improper communication between cellular and humoral adaptive immunity in long-COVID patients, which can result in immune dysregulation, inflammation, and the clinical symptoms associated with this debilitating condition [77]. Further, a distinct cytokine profile is linked to various factors observed during the critical phase of this illness, encompassing the initiation of interferon synthesis, the secretion of interleukin (ILs) 2 and 7, and the triggering of granulocyte activation and tumor necrosis factor (TNF) production [145]. The cytokine profile causes intravascular hyper-inflammation with changes in angiogenesis and coagulation [78]. Other studies have shown that SARS-CoV-2 can trigger secondary diseases associated with an immunosuppression profile, such as comorbid hypertension (COPD) [79] and chronic obstructive pulmonary disease [80]. Several reported studies indicated that individuals with pre-existing COPD tended to experience more severe outcomes when infected with SARS-CoV-2 [146]. Based on the available evidence, hypertension was frequently observed in individuals with COVID-19, but it did not independently contribute to the occurrence of SARS-CoV-2 infection or the progression of COVID-19 [79]. The summary of immunological symptoms can be seen in Table 1.

The COVID-19 infection significantly affected rare genetic disorders, as shown in Guillain–Barre syndrome (GBS). In the GBS disorder your “own immune system” attacks your nerve cells [147]. The first symptoms are usually weakness and tingling in the hands and feet. These sensations can quickly spread, eventually paralysing the whole body [147]. Several studies have confirmed that GBS also occurs in late-COVID patients [81,82,83]. One of the first studies explaining COVID-19 effects on GBS patients was published in 2020, where the authors revealed that two weeks after COVID-19 infection, quadriplegia and bilateral facial paresis occurred. Further, the electrodiagnostic findings of the patient demonstrated acute motor-sensory axonal polyneuropathy [82]. It is known that hypoxemic pneumonia generally emerges around two weeks post-infection, and a minority of children and young adults experience the onset of Multisystem Inflammatory Syndrome in Children (MIS-C) approximately four weeks after being infected by SARS-CoV-2. This condition shares characteristics with both Kawasaki disease and toxic shock syndrome mediated by superantigens [84]. Immunological examinations have unveiled hyper-inflammatory immune responses, differing from those observed in acute COVID-19 and Kawasaki disease. There is also evidence of T-cell activation, potentially triggered by a superantigen associated with SARS-CoV-19 [85]. A study conducted in 2023 indicated that among 2246, 151 patients harbored at least one ultra-rare variant of RTEL1 (regulator of telomere elongation helicase 1), chosen as a distinctive marker for acute severity in diseases correlated to liver functions causing lung fibrosis [148].

## 8. Long-COVID and the Endocrine System

An ACE2 receptor has been found in several endocrine glands, including the pancreas, thyroid gland, ovaries, and testes. The presence of the ACE2 receptor on the endocrine system suggests that SARS-CoV-2 may cause alternations in endocrine function. To date, several endocrine disorders such as diabetes, autoimmune thyroiditis, adrenal insufficiency, and male infertility have been reported in association with post-COVID-19 disease [86,87,88,89].

A study by Szczerbiński et al. [90] investigated the long-term effects of SARS-CoV-2 infection on the endocrine system in two distinct groups of patients. The first group of patients (*n* = 39) or case group was examined six months after COVID-19 and the control group consisted of age and sex-matched individuals before the COVID-19 pandemic. According to the obtained results, they found significantly higher levels of thyroid-stimulating hormone (TSH) and anti-thyroid peroxidase (aTPO) antibodies and lower free triiodothyronine (FT3) and FT4 levels in the case group compared to controls, indicating potential autoimmune hypothyroidism in patients recovered from COVID-19. A group of 334 COVID-19 patients who underwent TSH and FT4 measurements at admission showed normal levels and were euthyroid (86.5%). However, when these results were compared to non-COVID-19 cases (*n* = 122), a small reduction in TSH and FT4 levels was observed. Patients with COVID-19 had a reduction in TSH and FT4 levels when compared to their results from 2019, which was not observed in patients without COVID-19 [95]. COVID-19 genome-wide association studies were used to analyse the effect of COVID-19 on thyroid dysfunction, and the results identified that COVID-19 susceptibility and its severity might increase the risk of hypothyroidism [96].

It is well known that the extensive use of steroids can lead to steroid-induced diabetes mellitus [97]. Patients with severe COVID-19 typically receive an 8–10-day course of dexamethasone, with the possibility of prolonged steroid therapy for an extended duration. It has been observed that the prolonged use of steroids for several weeks to treat COVID-19, in some cases, has led to acute adrenal insufficiency [88]. Given that SARS-CoV-2 can be found in the adrenal glands of deceased COVID-19 patients and can infect adrenal cells in laboratory settings, it is reasonable to assume that COVID-19 may cause damage to the adrenal glands [88]. A significant cohort study utilising the US Department of Veterans Affairs national database assembled a group of 181,280 individuals who tested positive for COVID-19. The study found that, with a median follow-up period of 352 days, COVID-19 patients exhibited a heightened risk of developing diabetes compared to the control group [94].

Additionally, male patients recovered from COVID-19 had significantly lower levels of testosterone [90]. Evaluation of male reproductive health after three months from COVID-19 infection showed the presence of erectile dysfunction in one-third of tested subjects; however, this study showed that COVID-19 did not significantly affect testosterone levels, with only five subjects (6.2%) having testosterone levels below the laboratory reference range [91]. In a study involving 358 patients with COVID-19 and 92 patients without COVID-19, it was found that serum total testosterone levels were notably reduced in individuals with COVID-19 compared to those without the virus. Moreover, upon stratifying patients with COVID-19 based on disease severity, individuals with severe cases exhibited lower total testosterone levels in contrast to those with mild to moderate COVID-19 [92]. After a 7-month follow-up in men who recovered from COVID-19, overall testosterone increased in almost 90% of patients and further decreased in 10% of the entire cohort. However, 55% of men had subnormal testosterone levels suggestive of hypogonadism [93]. The summary of endocrine complications related to long-COVID is shown in Table 1.

## 9. Long-COVID and Gastrointestinal System

Studies have shown that patients with long-COVID are at greater risk of developing digestive diseases [149,150]. It is well known at this point that SARS-CoV-2 has adverse effects on different parts of the digestive system [98,105]. Epithelial cells of the gastrointestinal system (GI) have shown the most expression of ACE2 receptors in patients infected with COVID-19 [140].

It has been shown that the degree of GI symptoms in patients with severe COVID-19 was higher [150]. Also, the presence of GI symptoms has prolonged the duration of hospitalisation for infected patients compared to those who had not experienced any GI symptoms [151]. Accumulating evidence demonstrated that the SARS-CoV-2 genome was detected in the faeces of almost half of the infected patients over time [152], suggesting an interaction between the digestive system and the virus that could leave the patient with long-term sequelae [98,105,152,153,154,155].

Choudhury et al. have reported that 12% of patients after COVID-19 and 22% as part of long-COVID have shown GI symptoms [100]. Diarrhea, nausea, vomiting, loss of appetite, loss of taste, irritable bowel syndrome, and abdominal pain are the main common GI symptoms among patients in the early stages and even post-infection with the disease [98,99,100,105]. During the acute phase of the disease, GI symptoms were primarily diarrhea, nausea, and abdominal pain, respectively [98,101,102]. One study has observed that even 12 weeks after diagnosis, diarrhea was present [103].

Different studies have indicated a significant link between COVID-19 and elevation in the risk of developing digestive system disorders in the long term [104,150]. These conditions include acid-related disorders (dyspepsia, gastroesophageal reflux disease, and peptic ulcer disease), motility disorders, belching, hepatic and biliary disease, functional intestinal disorders, rectal bleeding, liver damage, acute pancreatitis, and cholangiopathy [99,104,105].

A meta-analysis performed by Cheung et al. [152], which includes 60 studies, demonstrates that, among 4243 patients, the commonness of GI symptoms was 17.6%, comprising 11.8% of non-severe and 17.1% severe cases of COVID-19 [152].

In another study, Weng et al. [99] investigated the prevalence of GI symptoms with long-COVID among 117 patients who were contacted 90 days after discharge. In total, 15 of these patients were diagnosed with GI symptoms on admission and 49 of them during hospitalisation. After 90 days, 52 of the patients reported GI symptoms. Among them, 15 patients had GI symptoms on admission day and during hospitalisation, 34 patients reported GI symptoms during hospitalisation, and 3 after discharge [99].

The GI tract is home to most human microbiota and it has been found that long-COVID could alter the population of this ecosystem [106,107]. Based on research studies, this alteration has mostly affected oral and gut microbiome even after recovery from COVID-19 [108,109,110]. Zhang et al. [156] have compared this effect in patients with and without GI symptoms of long-COVID. Patients who did not report GI symptoms showed little difference in the gut and oral microbiota compared to healthy controls. However, collected samples from patients with GI symptoms at baseline and the 3-month period confirmed significant differences and ectopic colonisation of the oral cavity by gut microbes. Their results have shown the expression of serum metabolites such as 4-chlorophenylacetic acid, 5-sulfoxymethylfurfural, and estradiol valerate with potential toxicity, which could be related to the differentially abundant bacteria including Neisseria, Lautropia, and Agrobacterium [156]. The summary of gastro-intestinal complications related to long-COVID is represented in Table 1.

## 10. Long-COVID and Dermatological Complications

Investigations have revealed an association between SARS-CoV-2 infection and skin changes in the post-infection stage [117,118]. However, this viral infection could result in various cutaneous clinical patterns throughout the disease [157]. Interestingly, accumulated evidence demonstrates that some of the cutaneous patterns are considered prodromal signs and could possibly represent an early indicator of infection or the only symptom of asymptomatic carriers [157,158].

More than 30 different skin eruptions caused by the COVID-19 virus have been described [159] and the characteristic manifestation of these dermatological conditions varies between individuals [117,118]. For instance, urticaria with itchy welts on the skin could be presented jointly with other COVID-19 symptoms [111,112]. Livedo reticularis, a condition that is characterised by mottled, purplish discoloration that forms a net-like pattern has also been reported, especially among infected patients with pre-existing vascular issues [113,114]. Other conditions such as petechiae and purpura have been found to be related to COVID-19 infection, particularly in severe cases, which could be due to coagulation abnormalities [112,114,115,116]. Also, there have been diagnoses of vesicular rash, with similarities to chickenpox or herpes infection, in some patients with COVID-19 [117,118]. Discoloration of the toes and fingers to red or purple, along with swelling and pain, are symptoms of COVID toe, which was found to be a rare condition present in infected patients [118]. Recent analysis has shown that the most commonly reported manifestations of COVID-19 are maculopapular rash, urticarial lesions, and chilblains [119].

Huang et al. studied 1655 hospitalised Chinese patients 6 months after infection onset and reported that only 47 patients (3%) had reported skin rashes [38]. In addition, other information on the long-term effects of COVID-19 has identified persistent cutaneous and extracutaneous symptoms, such as the absence of hair follicles, destruction of adipose tissue, and damage to the epidermal layer [120,121,122].

Sachdeva et al. [123] have conducted three case reports and a literature review on different manifestations in patients infected with SARS-CoV-2. The result of the study demonstrates that maculopapular exanthem (morbilliform) was reported by 36.1% of the patients, making it the most common cutaneous condition related to COVID-19, followed by papulovesicular rash (34.7%), urticaria (9.7%), painful acral red–purple papules (15.3%), livedo reticularis lesions (2.8%), and petechiae (1.4%) [123].

McMahon et el. [120] have investigated dermatological conditions in individuals from 41 countries who were laboratory-confirmed or suspected COVID-19 patients and cases who only were diagnosed with the skin condition. Their results showed that the median duration of morbilliform eruptions was 7 days. Urticaria lasted a median of 4 days, and the maximum duration of this condition was 28 days. Also, the longest median duration belonged to papulosquamous eruptions, which lasted 20 days, and one case was reported to last for 70 days [120].

Ica et al. [119] have published review research on skin lesions in COVID-19 patients in the short term and the long term. The study has not identified any lesions that were more likely to persist long term. Moreover, no pattern was detected to demonstrate if a chronic skin disease would be a result of long-COVID. However, perniosiform lesions were pointed out as the most evaluated lesions in the long term [119]. It also needs to be noted that as novel virus variants arise, new cutaneous patterns will be expected in long-COVID syndrome [159]. The summary of dermatological complications can be seen in Table 1.

## 11. Conclusions

This review provides an in-depth examination of the numerous complex health concerns related to the SARS-CoV-2 Post-acute Sequelae (PASC). This research emphasises the important public health consequences of persisting neuropsychiatric, cardiovascular, pulmonary, endocrine, gastrointestinal, and dermatological problems found in long-term COVID patients. The comprehensive analysis of these long-term impacts highlights the critical need for more research and integrative solutions to control and alleviate the issues faced by patients experiencing long-COVID.

PASC presents complex health issues with significant public health implications. Neuropsychiatric and neurological symptoms, such as cognitive impairment, sensory abnormalities, headaches, and sleep disturbances, are common, especially among hospitalised individuals and especially women. Respiratory SARS-CoV-2 infection leads to neurological inflammation, affecting neuronal functioning and cognitive abilities. Autoimmune responses, viral reactivation, and cerebrovascular issues contribute to these symptoms, alongside persistent cognitive impairments and mental health concerns such as anxiety and depression. Taking into consideration the impact of the virus, cardiovascular symptoms like chest pain, palpitations, and elevated heart rate are observed in long-term COVID-19 patients, with unclear pathophysiological links. Persistent hematological problems contribute to clotting disorders and coagulopathy, posing serious risks. Respiratory issues persist post-COVID-19, including chest discomfort, shortness of breath, and cough, with some experiencing severe respiratory failure and acute respiratory distress syndrome (ARDS). Kidney involvement raises the risk of sepsis and renal complications, including fibrosis. On the other hand, endocrine glands are affected, resulting in post-COVID diseases like diabetes, thyroiditis, and adrenal insufficiency. Long-COVID may also impact male reproductive health, with reduced testosterone levels and erectile problems post-recovery. The digestive system experiences symptoms post-infection, potentially leading to acid-related illnesses, motility problems, and pancreatitis. Changes in the oral and gut microbiota are noted among patients with GI symptoms, indicating potential risks. Skin abnormalities, ranging from urticaria to COVID toe, are common, with long-term consequences like hair loss and epidermal damage. However, no definitive patterns indicate chronic skin illnesses caused by long-COVID.

In summary, long-COVID poses a complicated set of health difficulties, mostly impacting the neurological, cardiovascular, pulmonary, endocrine, digestive, and dermatological systems, with significant consequences for public health. The persistence of neuropsychiatric symptoms, cardiovascular complications, respiratory issues, endocrine disruptions, gastrointestinal disturbances, and skin abnormalities emphasises the multifaceted nature of long-COVID. Nutritional treatment is critical for chronic disease, with the ability to reduce viral infection symptoms and influence epigenetic changes between populations [160]. High-fat diets can lower ACE2 expression, increasing vulnerability to SARS-CoV-2 [161]. Lifestyle variables such as nutrition and sleep patterns influence innate immune responses. Vitamin D deficiency has been linked to an increase in SARS-CoV-2 risk, and supplementation may provide protection against infection and respiratory diseases [162]. However, data promoting vitamin D for the long-term effects of COVID-19 are poor, despite the ability to reduce tiredness and weakness [163]. Probiotics, which are known to improve immunity and reduce inflammation by restoring gut health, have shown promise in lung and mental health. Lactobacillus plantarum has antiviral properties against SARS-CoV-2 in intestinal cells [164]. Though the advantages of probiotics are well recognised, their function in COVID-19 treatment is unknown [165]. However, more study is needed to better understand these relationships and optimise dietary methods, such as sunshine exposure and vitamin D-rich foods, for COVID-19 prevention and long-term risk reduction.

## Figures and Tables

**Figure 1 biomedicines-12-00913-f001:**
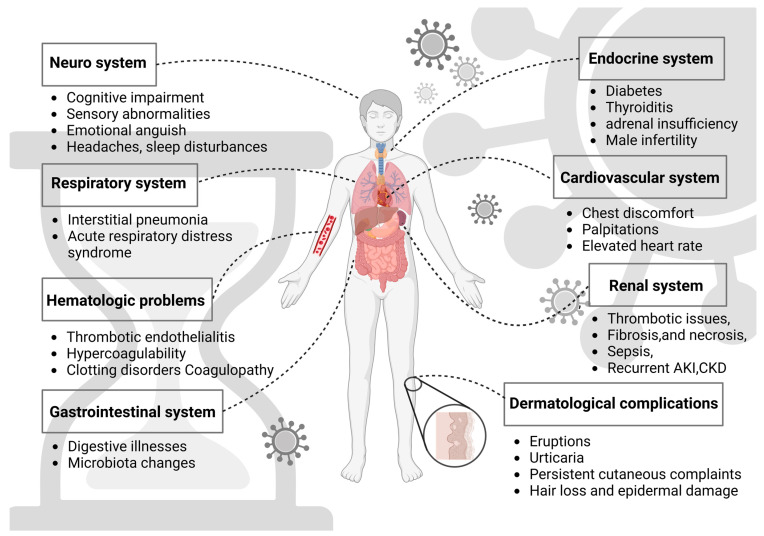
Graphical representation of organ systems affected by long-COVID. The figure was created with BioRender.

**Table 1 biomedicines-12-00913-t001:** Relationship between long-COVID and affected organ systems.

Organ System	Symptoms	Pathology	Refs.
Nervous system	Cognitive impairment (brain fog), memory issues, ageusia (loss of taste), anosmia (loss of smell), headaches, sleep disturbances, depression, anxiety		[23,24,26]
Cardiovascular system	Chest discomfort or tightness, palpitations, dizziness, rise in resting heart rate	Myocarditis, pericarditis, coronary artery disease	[47,48,49,50,51,52,53,54,55,56,57,58,59,60,61]
Haematologic problems		Persistent thrombotic endothelialitis, systemic hypercoagulability, fibrinaloid microclots, platelet hyperactivation, endothelial dysfunction, clotting disorders, SARS-CoV-2-mediated immune thrombocytopenia, persistent lymphocytopenia, coagulation	[59,60,61,62,63]
Respiratory system	Chest pain, breathlessness, cough, pulmonary dysfunction	Diffusion impairment, restrictive ventilatory defects, reduction in diffusion capacity	[64,65,66,67,68,69]
Renal system	Decreased renal perfusion, hypotension, ischaemia, salt diuresis, sympathetic stimulation, increased salt and water retention	Systemic inflammation, thrombotic issues, fibrosis, necrosis, sepsis, recurrent AKI, CKD	[70,71,72,73,74,75]
Immune complications		Systemic inflammation, immune dysregulation, debilitating condition, intravascular hyper-inflammation with changes in angiogenesis and coagulation, comorbid hypertension, chronic obstructive pulmonary disease, hypertension, Guillain–Barre syndrome, quadriplegia and bilateral facial paresis, acute motor-sensory axonal polyneuropathy, multisystem inflammatory, syndrome in children, hyper-inflammatory immune responses	[76,77,78,79,80,81,82,83,84,85]
Endocrine system		Diabetes, autoimmune thyroiditis, adrenal insufficiency, male infertility, autoimmune hypothyroidism, steroid-induced diabetes mellitus, acute adrenal insufficiency	[86,87,88,89,90,91,92,93,94,95,96,97]
Gastrointestinal system	Diarrhea, nausea, vomiting, loss of appetite, loss of taste, abdominal pain	Acid-related disorders (dyspepsia, gastroesophageal reflux disease, peptic ulcer disease), motility disorders, belching, hepatic and biliary disease, functional intestinal disorders, rectal bleeding, liver damage, acute pancreatitis, cholangiopathy, irritable bowel syndrome, microbiota alterations	[98,99,100,101,102,103,104,105,106,107,108,109,110]
Dermatological complications	Itchy welts		[111,112]
	Mottled, purplish discoloration that forms a net-like pattern	Livedo reticularis	[113,114]
		Petechiae and purpura	[112,114,115,116]
	Similarities to chickenpox or herpes infection	Vesicular rash	[117,118]
	Discoloration of the toes and fingers to red or purple, swelling, pain	COVID toe	[118]
		Maculopapular rash, urticarial lesions, chilblains	[119]
		Skin rashes	[38]
	Absence of hair follicles, destruction of adipose tissue, damage to the epidermal layer	Persistent cutaneous extracutaneous symptoms	[120,121,122]
	Livedo reticularis lesions, petechiae, painful acral red–purple papules	Maculopapular exanthem (morbilliform), papulovesicular rash, urticaria	[123]
		Morbilliform eruptions, urticaria, papulosquamous eruptions	[120]
		Pernio lesions	[119]

## Data Availability

The data are contained within the article.

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
