# Peer review of "The Aftermath of COVID-19: Exploring the Long-Term Effects on Organ Systems"

_biomedicines, 2024, doi:10.3390/biomedicines12040913_

Round 1

Reviewer 1 Report

Comments and Suggestions for Authors

The presented paper provides a detailed review of articles on the long-term effects of COVID for various organ systems. The authors of the paper did not do any research themselves. They simply reviewed the work of other authors. I have no critical comments for the presented review. In my opinion, the manuscript can be published.

Author Response

Comment: The presented paper provides a detailed review of articles on the long-term effects of COVID for various organ systems. The authors of the paper did not do any research themselves. They simply reviewed the work of other authors. I have no critical comments for the presented review. In my opinion, the manuscript can be published.

Answer: Thank you very much for your opinion.

Reviewer 2 Report

Comments and Suggestions for Authors

This paper reviews the emerging evidence and explains the interplay between various organ systems and long-term COVID-19, which is the post-acute sequelae of SARS-CoV-2 infection (PASC), indicating different physiological systems causing persistent symptoms. This paper is informative and instructive, and the topic is interesting. The paper merits publication after addressing the minor comments below.  

Minor comments:

1.      What is the major novelty of this paper?

2.      What is the scientific significance of this summary of the aftermath of Covid-19? I suggest the paper add a section to discuss mitigating the long Covid effects.

3.      Figure 1: I would point each organ system to the corresponding spot on the human body. 

Author Response

This paper reviews the emerging evidence and explains the interplay between various organ systems and long-term COVID-19, which is the post-acute sequelae of SARS-CoV-2 infection (PASC), indicating different physiological systems causing persistent symptoms. This paper is informative and instructive, and the topic is interesting. The paper merits publication after addressing the minor comments below. 

Answer: Thank you for your input.

Minor comments:

  1. What is the major novelty of this paper?

Answer: Thank you for this question. We added a paragraph discussing the novelty of this article in the conclusion section.

  1. What is the scientific significance of this summary of the aftermath of Covid-19? I suggest the paper add a section to discuss mitigating the long Covid effects.

Answer: Thank you for this valuable comment. We added a section at the end of the Conclusion part.

  1. Figure 1: I would point each organ system to the corresponding spot on the human body.

Answer: We appreciate this comment. The Figure 1 has been modified according to your input.